# Active Commuting to School and Physical Activity Levels among 11 to 16 Year-Old Adolescents from 63 Low- and Middle-Income Countries

**DOI:** 10.3390/ijerph17041276

**Published:** 2020-02-17

**Authors:** Miguel Peralta, Duarte Henriques-Neto, Joana Bordado, Nuno Loureiro, Susana Diz, Adilson Marques

**Affiliations:** 1CIPER, Faculdade de Motricidade Humana, Universidade de Lisboa, 1499-002 Lisboa; Portugal; mperalta@fmh.ulisboa.pt (M.P.); duarteneto@campus.ul.pt (D.H.-N.); 2ISAMB, Universidade de Lisboa, 1649-028 Lisboa, Portugal; nloureiro@ipbeja.pt; 3Faculdade de Motricidade Humana, Universidade de Lisboa, 1499-002 Lisboa; Portugal, sucris.diz@gmail.com (S.D.); 4Escola Superior de Educação, Instituto Politécnico de Beja, 7800-295 Beja, Portugal

**Keywords:** active travel, physical inactivity, region, school-aged children

## Abstract

Background: Global physical activity levels are low. Active commuting to school is a low-cost and sustainable behaviour that promotes adolescents’ physical activity levels. Despite its importance, data on low- and middle-income countries is scarce. This study aimed to assess the relationship between active commuting to school and physical activity (PA) levels among 11–16 years-old adolescents from 63 low- and middle-income countries and six world regions. Methods: Data were from the GSHS database. Participants were 187,934 adolescents (89,550 boys), aged 11–16 years-old, from 63 low- and middle-income countries. Active commuting to school and PA were self-reported as the number of days adolescents walked or cycled to school and engaged in physical activity for at least 60 min in the past 7 days. Results: Boys and girls who actively commuted to school presented higher prevalence of attaining the PA recommendations, but only for the 13–14 (boys: 16.6% versus 22.0%; girls: 9.8% versus 14.6%) and 15–16 (boys: 16.3% versus 21.6%; girls: 8.0% versus 14.0%) year-old age groups. Only for Oceania, Central Asian, Middle Eastern, and North African girls and Sub-Saharan African boys no difference was found in the prevalence of attaining the PA recommendations between those who actively commuted to school and those who did not. Boys who actively commuted to school were 42% (95% CI: 1.37, 1.46) more likely to achieve the PA recommendations, while girls were 66% (95% CI: 1.59, 1.73) more likely to achieve the PA recommendations. Conclusions: Active commuting to school is associated with the adolescents’ physical activity levels. However, it may have a lesser influence in helping younger adolescents attaining physical activity recommendations. Public health authorities should promote active commuting to school among adolescents in order to improve the PA levels and promote health.

## 1. Introduction

Global physical activity levels are low. Even though the World Health Organization (WHO) recommends adolescents to engage in at least 60 min of daily moderate-to-vigorous physical activity [1], it is estimated that 77.6% of boys and 84.7% of girls, aged 11 to 17 years old are physically inactive [2]. This is a worrying fact, as physical activity is associated with several health benefits, including cardiovascular, bone, metabolic, and mental health [3].

There are several strategies with the potential to increase adolescents’ physical activity levels. Active commuting to school (e.g., walking or cycling) is one of them [4]. It is a low-cost and sustainable behaviour that several studies suggest to be an effective strategy to increase adolescents overall physical activity [5,6]. Active commuting to school by walking contributes to increased daily physical activity in both boys and girls [4]. Findings from randomized control trials have demonstrated that those in the active commuting walking group (experimental group) increased the number of minutes in moderate-to-vigorous physical activity [7,8]. Similarly, those who cycle to school are more active than those who do not [9]. Whether assessing physical activity through a questionnaire, accelerometer or pedometer, studies have observed that active commuting to school is positively associated with physical activity. This relationship is further supported by evidence demonstrating that active commuters have higher physical activity levels on weekdays [10,11]. Furthermore, among adolescents living in urban contexts, it was found that more than half of their daily physical activity was due to active commuting [12].

As an intervention strategy to promote physical activity, a recent systematic review concluded that active commuting is effective [6]. Furthermore, from an educational perspective, active transportation stimulated at early ages [10] can help to consolidate this behaviour throughout life [13]. However, despite the effectiveness of active commuting as a strategy to promote physical activity, the number of adolescents active commuting to school in high-income countries has been decreasing [14,15,16]. Among low- and middle-income countries, data are limited. Notwithstanding, investigations from Brazil [17], Mozambique [18] and Vietnam [19] have shown that in these countries a downward trend in active commuting to school is also observed.

Although there is strong evidence for the relationship between active commuting to school and physical activity levels in adolescents, the majority of the studies were conducted in high-income countries. Thus, in low- and middle-income countries, data on this relationship is limited [20]. International studies using comparable methods are important to establish trends or generalization of the associations, support decision making and inform global efforts on the promotion of physical activity and sustainable transport. For that reason, and from a public health perspective, more data about active commuting on low- and middle-income countries is required. This information should allow to better support public health policies and decision making regarding the promotion of physical activity in these countries. Therefore, the aim of this study was to assess the relationship between active commuting to school and physical activity levels among 11–16 years-old adolescents from 63 low- and middle-income countries and six world regions.

## 2. Materials and Methods

### 2.1. Participants and Procedures

This study is based on the Global School-Based Student Health Survey (GSHS), a cross-sectional survey aimed primarily at students aged between 13 and 17 years old from several low- and middle-income countries worldwide. The GSHS uses a standardized scientific sample selection process, common school-based methodology and self-administered questionnaire supported by the WHO with the collaboration of the United Nations’ UNICEF, UNESCO, and UNAIDS, and the technical assistance from the Centers for Disease Control and Prevention. The questionnaire includes validated survey items selected from 10 core modules, including: alcohol use, dietary behaviours, drug use, hygiene, mental health, physical activity, protective factors, sexual behaviours, tobacco use, and violence and unintentional injury. More information about the GSHS can be found elsewhere [21].

Publicly available GSHS data, collected from 2007 to 2015, for low- and middle-income countries was used. Data collected in the surveys were self-reported. For this study, adolescents aged between 11 and 16 years-old were considered, as this was the age range that guaranteed participants in each age from all the available countries. Furthermore, only adolescents who reported data on sex, active commuting to school, and physical activity were considered. The final sample consisted of 187,934 adolescents (89,550 boys, 98,384 girls), aged 11 to 16 years-old (mean age 14.3 ± 1.2), from 63 countries (1841 Afghanistan, 4350 Algeria, 1177 Antiqua and Barbuda, 26,636 Argentina, 1299 Bahamas, 2857 Bangladesh, 1548 Barbados, 1993 Belize, 3378 Bolivia, 1616 British Virgin Islands, 2283 Brunei Darussalam, 2313 Cambodia, 1591 Chile, 2622 Costa Rica, 1735 Curacao, 1512 Dominica, 2424 Egypt, 1833 El Salvador, 1590 Fiji, 1763 Ghana, 3866 Guatemala, 2269 Guyana, 1696 Honduras, 1944 Iraq, 1551 Kiribati, 2460 Kuwait, 2522 Laos, 2162 Lebanon, 20672 Malaysia, 1921 Mauritania, 2135 Mauritius, 2744 Myanmar, 4431 Mongolia, 2760 Morocco, 1015 Mozambique, 2604 Namibia, 477 Nauru, 137 Niue, 2360 Oman, 5054 Pakistan, 4281 Palestinian territory, 2811 Peru, 7730 Philippines, 1675 Saint Kitts and Nevis, 2067 Samoa, 2389 Seychelles, 1262 Solomon Islands, 2040 Sudan, 1620 Suriname, 3027 Syria, 3193 Tanzania, 4745 Thailand, 2241 Timor-Leste, 110 Tokelau, 2148 Tonga, 2639 Trinidad and Tobago, 865 Tuvalu, 2461 United Arab Emirates, 3415 Uruguay, 1080 Vanuatu, 2267 Vietnam, 855 Wallis and Futuna, and 1872 Yemen) and six world regions (Central Asia, Middle East, North Africa, East and Southeast Asia, Latin America and Caribbean, Oceania, South Asia, and Sub-Saharan Africa) defined by the World Bank as low- and middle-income countries at the time of data collection.

### 2.2. Measures

Participants filled a self-administrated questionnaire during a school lesson (questionnaires were country-specific and translated to the country official language). Active commuting to school was self-reported as the number of days adolescents walked or cycled to school in the past week (‘During the past 7 days, on how many days did you walk or ride a bicycle to or from school?’). Those who reported active commuting at least 1 day in the past week were considered to active commute to school. Adolescents were asked to report the number of days they engaged in physical activity for at least 60 min in the past 7 days (‘During the past 7 days, on how many days were you physically active for a total of at least 60 min per day?’). Answers were given on an 8-point scale (0 = none to 7 = daily). This has been shown to be a valid and reliable question for assessing adolescents’ physical activity in epidemiological research [22].

### 2.3. Data Analysis

The mean frequency of physical activity (days/week) and the prevalence of attaining the physical activity recommendations for those who actively commuted to school and those who did not, according to age group (11–12, 13–14 and 15–16 years old) and region (Central Asia, Middle East, and north Africa, East and southeast Asia, Latin America and Caribbean, Oceania, South Asia, Sub-Saharan Africa) were calculated and the 95% confidence intervals (CI) were estimated. Logistic regression was performed to analyse the relationship between active commuting to school and attaining the physical activity recommendations. Unadjusted and adjusted analyses for age and region were tested. Attaining the physical activity recommendations was defined, using the WHO criteria, as engaging in moderate-to-vigorous physical activity for at least 60 min every day [1]. All analyses were stratified by sex. Data analysis was performed using IBM SPSS Statistics version 25 (IBM Corp., Armonk, NY, USA).

## 3. Results

Participants’ characteristics are presented in Table 1. Most adolescents were aged between 13 and 15 years old and only 15.6% (19.5% boys, 12.0% girls) engaged in at least 60 min of physical activity every day. More than half of the adolescents (56.1%) active commuted to school.

In both boys and girls, active commuting to school was related to the physical activity levels (Table 2). For every age group, those who actively commuted to school had a higher frequency of physical activity (days/week). The oldest age group showed the biggest (boys, 0.61 days/week, 95% CI: 0.56, 0.61; girls, 0.66 days/week, 95% CI: 0.62, 0.70) mean difference, whereas the youngest age group presented the lowest (boys, 0.47 days/week, 95% CI: 0.34, 0.60; girls, 0.42 days/week, 95% CI: 0.31, 0.53). Furthermore, those who actively commuted to school presented higher prevalence of attaining the physical activity recommendations, but only for the 13–14 (boys: 16.6% versus 22.0%; girls: 9.8% versus 14.6%) and 15–16 (boys: 16.3% versus 21.6%; girls: 8.0% versus 14.0%) year-old age groups.

The relationship between active commuting to school and physical activity according to region is shown in Table 3. For every region, boys and girls that active commuted to school had higher frequency of physical activity. South Asia (boys, 1.05 days/week, 95% CI: 0.91, 1.19; girls, 1.01 days/week, 95% CI: 0.86, 1.01) and Oceania (boys, 0.74 days/week, 95% CI: 0.60, 0.88; girls, 0.92 days/week, 95% CI: 0,80, 1,04) were the regions with the biggest mean difference in the frequency of physical activity between those who actively commuted to school and those who did not, while Central Asia, Middle East, and North Africa (boys, 0.43 days/week, 95% CI: 0.35, 0.51; girls, 0.29 days/week, 95% CI: 0.22, 0.36) had the lowest. Only for Oceania, Central Asian, Middle Eastern, and North African girls and Sub-Saharan African boys no difference was found in the prevalence of attaining the physical activity recommendations between those who actively commuted to school and those who did not.

Table 4 presents the odds ratio for attaining the physical activity recommendations according to active commuting to school. In the adjusted analysis, boys who actively commuted to school were 42% (95% CI: 1.37, 1.46) more likely to achieve the physical activity recommendations, and girls who actively commuted to school were 66% (95% CI: 1.59, 1.73) more likely to achieve the physical activity recommendations.

## 4. Discussion

This study aimed to assess the relationship between active commuting to school and the physical activity levels of 11–16 year-olds from 63 low- and middle-income countries and six world regions. It was found that both boys and girls who actively commuted to school engaged more frequently in physical activity and were more likely to attain the WHO physical activity recommendations. However, age and region differences were found.

Physical activity is an important health behaviour associated to several health benefits [3]. A previous systematic review found that adolescents who actively commuted to school are more physically active and that active school travel interventions often lead to increases in adolescents’ physical activity levels [5] and can help to consolidate this behaviour throughout life [13]. In accordance with previous results from mainly high-income countries, this study showed that in low- and middle-income countries, active commuting to school is also associated to physical activity, and that boys and girls who actively commuted to school were, respectively, 42% and 66% more likely to attain the physical activity recommendations. This is a relevant finding, as active commuting to school is an effective strategy to increase the globally low physical activity levels [2,5]. Furthermore, active commuting to school is also associated with lower measures of adiposity and better cardiorespiratory fitness [5,20]. Globally, international, regional and national public health authorities should promote active commuting to school among adolescents in order to improve the physical activity levels and promote health.

Although boys and girls from all age groups (11–12, 13–14 and 15–16 years-old) who actively commuted to school engaged more frequently in physical activity, the prevalence of attaining the physical activity recommendations was only higher for those who actively commuted in the 13–14 and 15–16 years-old age groups. This may be due to several reasons. On the one hand, the difference in the prevalence of attaining the physical activity recommendations between active commuters and non-active commuters was the smallest in the youngest age group. This suggests that for the 11–12 years-old age group, active commuting to school may not be a significant contributor to achieve the recommended physical activity levels, as it is for the 13–14 and 15–16 years-old age groups. On the other hand, the 11–12 years-old sample was much smaller than the other age groups, which may have led to less accuracy in determining significant statistical differences. Notwithstanding, physical activity levels are known to decrease across adolescence, thus younger adolescents are more physically active than their older peers [23,24]. Because of this, active commuting to school may have a lesser influence in helping younger adolescents attain physical activity recommendations, although it contributes to the overall physical activity levels.

A systematic review, performed mainly with studies from high-income countries in Europe, North America and Oceania, indicated that active commuting to school is positively associated with the physical activity levels of school-aged children and adolescents [5]. In this study, with data from low- and middle-income countries, similar findings were observed. However, the prevalence of boys and girls from Oceania; girls from Central Asia, Middle East, and North Africa; and boys from Sub-Saharan Africa attaining the physical activity recommendations was similar for active commuters and non-active commuters. These findings show that regional differences in the relationship between active commuting and physical activity may exist. Future research should try to explore possible reasons for this. Nonetheless, in all regions, boys and girls who actively commuted engaged more frequently in physical activity. Therefore, active commuting to school should be used as a strategy to promote physical activity across all regions.

The interpretation of this study’s findings warrants some public health policy implications for low- and middle-income countries. Active commuting to school seems to contribute to the overall physical activity levels of adolescents from low- and middle-income countries, as it does for high-income countries. Taking into account the global levels of insufficient physical activity [2], public health authorities from these countries are recommended to use active commuting to school as a strategy to promote physical activity and health among adolescents. Additionally, the contribution of active commuting to school for achieving the physical activity recommendations appears to be more relevant for adolescents aged 13 and over. Thus, active commuting may be a more suitable strategy to promote physical activity among adolescents aged 13 and older than for their younger peers.

The present study has several limitations that are important to mention. Data was collected at the school setting, resulting in the exclusion of adolescents that do not attend schools. Also, data represented only adolescents aged 11–16 years-old and the GSHS focuses mainly on adolescents aged 13–17 years. Thus, the 11–12 years-old sample was small, in comparison to the other age groups, and adolescents aged 17 were not included. Data on active commuting to school and physical activity were self-reported and therefore subject to bias. Nevertheless, because of the large samples used in each national or regional survey, self-report was the most feasible methodology and is still the backbone of surveillance studies [25]. Furthermore, intensity is an important part of the WHO physical activity recommendations, as children and adolescents should engage in at least 60 moderate-to-vigorous intensity physical activity daily. However, in the GSHS questionnaire, used in this study, the question regarding physical activity does not consider intensity (only duration and frequency). This limitation could lead to bias in the classification of adolescents attaining the physical activity recommendations. There is still no consensus regarding the optimal cut-off criteria for dichotomizing active commuting [26]. In this study, a cut-off value of 1 day was used as the aim of the study was to compare those who did not actively commute to school (i.e., 0 days of active commuting) and those who did, even if only on 1 day. Additionally, because this study used a sample of 63 countries with different cultural and administrative settings, the number of days adolescents went to school could vary between countries. Thus, establishing a cut-off value different than one could lead to bias. Also, the question regarding active commuting does not allow to differentiate the type of active commuting, such as walking or cycling. It would be interesting to compare whether walking and cycling have similar or different contributions for the physical activity levels.

## 5. Conclusions

Active commuting to school is associated with the adolescents’ physical activity levels. However, some sex, age and regional differences were observed. Active commuting seems to have a greater effect among girls, as girls who actively commuted to school were 66% more likely to attain the physical activity recommendations, while boys were only 42%. Also, active commuting to school may have a lesser influence in helping younger adolescents attain the physical activity recommendations, as no differences were found in the 11–12 years-old group, although it contributes to the overall physical activity levels. Therefore, public health authorities should promote active commuting to school among adolescents in order to improve the physical activity levels and promote health, especially among adolescents aged 13 years old and over.

## Figures and Tables

**Table 1 ijerph-17-01276-t001:** Adolescents characteristics, including age, region, physical activity, and active commuting to school, by sex.

Adolescents’ Characteristics	Total (*n*= 187,934)		Boys (*n* = 89,550)		Girls (*n* = 98,384)	
	*n*	% (95% CI)	*n*	% (95% CI)	*n*	% (95% CI)
Age (years)						
11	1843	1.0 (0.5, 1.4)	827	0.9 (0.3, 1.6)	1016	1.0 (0.4, 1.7)
12	11,296	6.0 (5.6, 6.4)	4965	5.5 (4.9, 6.2)	6331	6.4 (5.6, 7.0)
13	39,403	21.0 (20.6, 21.4)	18,160	20.3 (19.7, 20.9)	21,243	21.6 (21.0, 22.1)
14	51,392	27.3 (27.0, 27.7)	24,488	27.3 (26.8, 27.9)	26,904	27.3 (26.8, 27.9)
15	49,720	26.5 (26.1, 26.8)	24,132	26.9 (26.4, 27.5)	25,588	26.0 (25.5, 26.5)
16	34,280	18.2 (17.8, 18.6)	16,978	19.0 (18.4, 19.5)	17,302	17.6 (17.0, 18.2)
Region						
Central Asia, Middle East, and north Africa	34,532	18.4 (18.0, 18.8)	16,243	18.1 (17.5, 18.7)	18,289	18.6 (18.0, 19.2)
East and southeast Asia	44,773	23.8 (23.4, 24.2)	21,158	23.6 (23.1, 24.2)	23,615	24.0 (23.5, 24.5)
Latin America and Caribbean	66,931	35.6 (35.3, 36.0)	31,938	35.7 (35.1, 36.2)	34,993	35.6 (35.1, 36.1)
Oceania	12,142	6.5 (6.0, 6.5)	5450	6.1 (5.5, 6.7)	6692	6.8 (6.2, 7.4)
South Asia	12,496	6.6 (6.2, 7.1)	7016	7.8 (7.2, 8.5)	5480	5.6 (5.0, 6.2)
Sub-Saharan Africa	17,060	9.1 (8.6, 9.5)	7745	8.6 (8.0, 9.3)	9315	9.5 (8.9, 10.1)
Physical activity (days/week)						
0	47,611	25.3 (24.9, 25.7)	20,574	23.0 (22.4, 23.5)	27,037	27.5 (26.9, 28.0)
1	38,660	20.6 (20.2, 21.0)	16,411	18.3 (17.7, 18.9)	22,249	22.6 (22.1, 23.2)
2	26,817	14.3 (13.9, 14.7)	11,617	13.0 (12.4, 13.6)	15,200	15.4 (14.9, 16.0)
3	18,642	9.9 (9.5, 10.3)	9036	10.1 (9.5, 10.7)	9606	9.8 (9.2, 10.4)
4	10,898	5.8 (5.4, 6.2)	5792	6.5 (5.8, 7.1)	5106	5.2 (4.6, 5.8)
5	10,630	5.7 (5.2, 6.1)	5602	6.3 (5.6, 6.9)	5028	5.1 (4.5, 5.7)
6	5360	2.9 (2.4, 3.3)	3033	3.4 (2.7, 4.0)	2327	2.4 (1.7, 3.0)
7	29,316	15.6 (15.2, 16.0)	17,485	19.5 (18.9, 20.1)	11,831	12.0 (11.4, 12.6)
Active commuting to school						
No	82,423	43.9 (43.5, 44.2)	37,380	41.7 (41.2, 42.2)	45,043	45.8 (45.3, 46.2)
Yes	10,5511	56.1 (55.8, 56.4)	52,170	58.3 (57.8, 58.7)	53,341	54.2 (53.8, 54.6)

CI: confidence interval.

**Table 2 ijerph-17-01276-t002:** Mean frequency of physical activity (day/week) and the prevalence of attaining the physical recommendations according to active commuting by age group.

**Active Commuting to School**	**Physical Activity (Days/Week)** **Mean (95% CI)**		
	**11–12 year-olds**	**13–14 year-olds**	**15–16 year-olds**
Boys			
No	2.43 (2.34, 2.53)	2.51 (2.48, 2.55)	2.59 (2.55, 2.63)
Yes	2.90 (2.80, 2.99)	3.10 (3.07, 3.13)	3.20 (3.16, 3.23)
Girls			
No	1.98 (1.90, 2.05)	2.00 (1.98, 2.03)	1.88 (1.85, 1.91)
Yes	2.40 (2.32, 2.48)	2.58 (2.55, 2.61)	2.54 (2.51, 2.56)
**Active Commuting to School**	**% Attaining Physical Activity Recommendations** **% (95% CI)**		
	**11–12 year-olds**	**13–14 year-olds**	**15–16 year-olds**
Boys			
No	16.7 (13.3, 20.0)	16.6 (15.2, 17.9)	16.3 (14.9, 17.7)
Yes	21.1 (17.9, 24.3)	22.0 (20.9, 23.1)	21.6 (20.5, 22.7)
Girls			
No	12.0 (9.0, 15.0)	9.8 (8.6, 11.0)	8.0 (6.6, 9.4)
Yes	14.9 (11.9, 18.0)	14.6 (13.5, 15.8)	14.0 (12.8, 15.2)

CI: confidence interval.

**Table 3 ijerph-17-01276-t003:** Mean frequency of physical activity (days/week) and the prevalence of attaining the physical activity recommendations according to active commuting by regions.

**Active Commuting to School**	**Physical Activity (Days/Week)** **Mean (95% CI)**					
	**Central Asia, Middle East, and north Africa**	**East and southeast Asia**	**Latin America and Caribbean**	**Oceania**	**South Asia**	**Sub-Saharan Africa**
Boys						
No	2.71 (2.65, 2.77)	2.34 (2.29, 2.39)	2.82 (2.78, 2.87)	2.29 (2.19, 2.39)	1.66 (1.56, 1.76)	2.36 (2.27, 2.44)
Yes	3.14 (3.09, 3.19)	2.95 (2.90, 3.00)	3.43 (3.39, 3.46)	3.03 (2.94, 3.12)	2.71 (2.63, 2.78)	2.90 (2.82, 2.99)
Girls						
No	2.01 (1.97, 2.06)	1.72 (1.68, 1.75)	2.13 (2.10, 2.17)	1.98 (1.90, 2.06)	1.94 (1.83, 2.06)	1.81 (1.74, 1.87)
Yes	2.30 (2.25, 2.35)	2.23 (2.19, 2.26)	2.75 (2.72, 2.78)	2.90 (2.82, 2.99)	2.95 (2.86, 3.04)	2.45 (2.38, 2.53)
**Active Commuting to School**	**% Attaining Physical Activity Recommendations** **% (95% CI)**					
	**Central Asia, Middle East, and north Africa**	**East and southeast Asia**	**Latin America and Caribbean**	**Oceania**	**South Asia**	**Sub-Saharan Africa**
Boys						
No	19.2 (17.1, 21.3)	14.3 (12.5, 16.2)	18.4 (16.8, 20.0)	13.9 (10.2, 17.7)	8.1 (3.7, 12.5)	15.6 (12.6, 18.6)
Yes	23.3 (21.5, 25.1)	20.4 (18.8, 22.1)	23.4 (22.2, 24.7)	17.0 (13.8, 20.2)	19.4 (16.9, 21.8)	20.7 (18.0, 23.5)
Girls						
No	10.9 (8.9, 12.9)	5.5 (3.8, 7.3)	10.1 (8.6, 11.6)	10.9 (7.6, 14.1)	12.5 (8.3, 16.7)	9.9 (7.2, 12.5)
Yes	13.3 (11.5, 15.2)	10.1 (8.4, 11.7)	15.5 (14.3, 16.8)	15.9 (12.9, 18.9)	21.7 (18.8, 24.6)	16.2 (13.4, 18.9)

CI: confidence interval.

**Table 4 ijerph-17-01276-t004:** Odds ratio for attaining the physical activity recommendations according to active commuting to school.

Active Commuting to School	Attaining Physical Activity RecommendationsOR (95% CI)			
	Boys		Girls	
	Unadjusted	Adjusted	Unadjusted	Adjusted
No	1.00 (ref)	1.00 (ref)	1.00 (ref)	1.00 (ref)
Yes	1.41 (1.36, 1.46)	1.42 (1.37, 1.47)	1.65 (1.58, 1.71)	1.66 (1.59, 1.73)

OR: odds ratio; CI: confidence interval. Analysis adjusted for age and region.

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
