# Peer review of "Active Commuting to School and Physical Activity Levels among 11 to 16 Year-Old Adolescents from 63 Low- and Middle-Income Countries"

_ijerph, 2020, doi:10.3390/ijerph17041276_

Round 1

Reviewer 1 Report

The research is based on an extensive survey that covers vast population of medium- and low-income counteries. Several issues need to be addressed before publication:

Introduction is rather thin. Please add more relevant papers and identify the knowledge gaps in the extant literature. Why the data of low- and middle-income countries is required? The contribution and significance of this work shall be emphasized.

A proper literature review is desirable, because there are plenty of publications related to this topic.

The detailed information about the survey can be given. For examples, the number of the respondents in each included country, the layout of the survey form, and the surveyed period.  

The robustness of the findings can be tested. For instance, the authors might use sectional samples to test their theory and the outcomes of the whole sample.

Based on the results, policy implications can be offered to the administration of these low- and middle-income countries.

Author Response

We appreciate the thoughtful comments and have modified the manuscript in response to the suggestions, which we think will further improve the quality of the manuscript.

Please find point-by-point response to the comments below. All changes in the manuscript are highlighted by using the track mode tool of the MS word.

On the behalf of all authors

Comment: The research is based on an extensive survey that covers vast population of medium- and low-income countries. Several issues need to be addressed before publication:

Response: Thank you for the comments. The issues stated were addressed.

Comment: Introduction is rather thin. Please add more relevant papers and identify the knowledge gaps in the extant literature. Why the data of low- and middle-income countries is required? The contribution and significance of this work shall be emphasized. A proper literature review is desirable, because there are plenty of publications related to this topic.

Response: The introduction has been improved, introducing more studies and better supporting the rational of the study.

Comment: The detailed information about the survey can be given. For examples, the number of the respondents in each included country, the layout of the survey form, and the surveyed period.

Response: The required information was added to the methods section.

Comment: The robustness of the findings can be tested. For instance, the authors might use sectional samples to test their theory and the outcomes of the whole sample.

Response: Thanks for the observation. Prior to the analysis, we performed random permutation test using the sample split method. The results between subsamples were similar.

Comment: Based on the results, policy implications can be offered to the administration of these low- and middle-income countries.

Response: Thank you. Public health policy implications of the findings were added to the discussion and conclusion.

Reviewer 2 Report

Good work

Author Response

Comment: Good work

Response: Thank you.

Reviewer 3 Report

Abstract
Include a background before the objectives.
Rewrite the conclusion based on the objectives and the results obtained.
Include that questionnaire or means of obtaining self-reported data participants.

Introduction
Line 35. Include reference: https://www.ncbi.nlm.nih.gov/pubmed/31968634
They should better justify the objectives of the study. It is very general.

Material and Methods
The section is very loose. They should give more information on how everything was done.
Explain how participants access the survey. How they are recruited?
Explain well how was the questionnaire that participants should answer. How were the questions. Perhaps you can use as a reference how to make the material and methods of the following reference https://www.ncbi.nlm.nih.gov/pubmed/31968634
Why did they decide that Those who reported active commuting at least one day in the past 76 week were considered to active commute to school? Any reference that supports it? A few days it seems to me to be considered active commute.
When talking about physical activity is moderate, vigorous, light. This is very important to determine the relationship you are looking for. WHO explains: Children and youth aged 5–17 should accumulate at least 60 minutes of moderate-to vigorous-intensity physical activity daily. Therefore, relating it to any type of activity would not be correct.

It would have been interesting if they were walking or cycling to school in order to determine social policies.

Author Response

We appreciate the thoughtful comments and have modified the manuscript in response to the suggestions, which we think will further improve the quality of the manuscript.

Please find point-by-point response to the comments below. All changes in the manuscript are highlighted by using the track mode tool of the MS word.

Comment: (Abstract) Include a background before the objectives.

Response: Added.

Comment: (Abstract) Rewrite the conclusion based on the objectives and the results obtained.

Response: two sentences directly related to the study findings were added to the conclusion.

Comment: (Abstract) Include that questionnaire or means of obtaining self-reported data participants.

Response: Added.

Comment: (Introduction) Line 35. Include reference: https://www.ncbi.nlm.nih.gov/pubmed/31968634

Response: The reference was included.

Comment: (Introduction) They should better justify the objectives of the study. It is very general.

Response: The objective of the study was better justified, stating the need for more data on low- and middle-income countries to inform public health policies and decision making.

Comment: (Material and Methods) The section is very loose. They should give more information on how everything was done.

Response: The section was improved, including more data on the GSHS.

Comment: (Material and Methods) Explain how participants access the survey. How they are recruited?

Response: The GSHS uses a standardized scientific sample selection process and common school-based methodology. Although it may vary from country to country, the typical procedure is the following: schools serving students from the desired ages are selected by the coordinator; parents were informed of the survey via a letter sent home one week prior to its administration at their child’s school, and were given the opportunity to opt out; survey coordinators explained the procedures to children in their classroom, and any child not willing to participate was excluded. A resumed version of this explanation was added to the methods section.

Comment: (Material and Methods) Explain well how was the questionnaire that participants should answer. How were the questions. Perhaps you can use as a reference how to make the material and methods of the following reference https://www.ncbi.nlm.nih.gov/pubmed/31968634

Response: The required information was added to the methods section.

Comment: (Material and Methods) Why did they decide that Those who reported active commuting at least one day in the past week were considered to active commute to school? Any reference that supports it? A few days it seems to me to be considered active commute.

Response: Thank you for your comment. To the best of our knowledge there is no cut-off for active commuting to school and studies are very heterogeneous. A recent study (Zaragoza et al. Active or Passive Commuter? Discrepancies in Cut-off Criteria among Adolescents. Int J Environ Res Public Health. 2019; 16(20): 3796) tried to bring some clarity to this matter, concluding that there is still no consensus regarding the optimal cut-off criteria for dichotomizing active versus passive commuters and that the context of the study should be taken into account.

Although most studies use a cut-off of half or more than half of weekdays / trips to classify adolescents has active commuters, the aim of the study was to compare those who did not active commuted to school (i.e., 0 days of active commuting) and those who did, even if only one day, to see the contribution of active commuting to the physical activity levels. Furthermore, this study includes several countries from different world regions and with different cultural and administrative settings. This means that the number of school days varies between countries, thus choosing a cut-off value different that 1 (for instance 2 or 3) could lead to misclassification of adolescents that go to school very few days a week. Taking into account the aim of the study and the cultural and administrative settings of each country we decided to classify as active commuters those who reported to active commuting to school at least one day in the past week.

The information explaining the use of the 1 day cut-off value and the inconsistence regarding the optimal cut-off criteria for dichotomizing active commuting was added to the limitation section.

Comment: (Material and Methods) When talking about physical activity is moderate, vigorous, light. This is very important to determine the relationship you are looking for. WHO explains: Children and youth aged 5–17 should accumulate at least 60 minutes of moderate-to vigorous-intensity physical activity daily. Therefore, relating it to any type of activity would not be correct.

Response: Thank you for your comment. In fact, you are correct. Unfortunately, the way the question about physical activity participation was asked does not allow to determine intensity (only duration and frequency). This issue is now properly addressed in the limitations section.

Comment: It would have been interesting if they were walking or cycling to school in order to determine social policies.

Response: Indeed, it would be interesting to have data on cycling and walking separately in order to compare different types of active commuting. However, data from the GSHS does not differentiates the type of active commuting. This was added to the limitations.

Round 2

Reviewer 1 Report

As the issues have been addressed, I recommend that the paper can be published.

Reviewer 3 Report

The authors have responded all my concerns. I think the manuscript has been improve. Therefore, The manuscript could be accepted if the editor considers.